# MMG-VL: A Vision-Language Driven Approach for Multi-Person Motion Generation

## Abstract

Generating realistic 3D human motion is crucial in the frontier applications of embodied intelligence, such as human-computer interaction and virtual reality. However, existing methods that rely solely on text or initial human pose inputs struggle to capture the rich semantic understanding and interaction with the environment, and most focus on single-person motion generation, neglecting the needs of multi-person scenarios. To address these challenges, we propose the **VL2Motion** generation paradigm, which combines natural language instruction and environmental visual inputs to generate realistic 3D human motion. The visual inputs not only provide precise analysis of spatial layouts and environmental details but also incorporate inherent 3D spatial and world knowledge constraints to ensure that the generated motions are natural and contextually appropriate in real-world scenarios. Building on this, we introduce **MMG-VL**, a novel **M**ulti-person **M**otion **G**eneration approach driven by **V**ision and **L**anguage for generating 3D human motion in multi-room home scenarios. This approach employs a two-stage pipeline: first, it uses *Vision-Language Auxiliary Instruction (VLAI)* module to integrate multimodal input information and generate multi-human motion instructions that align with real-world constraints; second, it utilizes *Scenario-Interaction Diffusion (SID)* module to accurately generate multiple human motions. Our experiments demonstrate the superiority of the VL2Motion paradigm in environmental perception and interaction, as well as the effectiveness of MMG-VL in generating multi-human motions in multi-room home scenarios. Additionally, we have released a complementary **HumanVL** dataset, containing 584 multi-room household images and 35,622 human motion samples, aiming to further advance innovation and development in this domain.

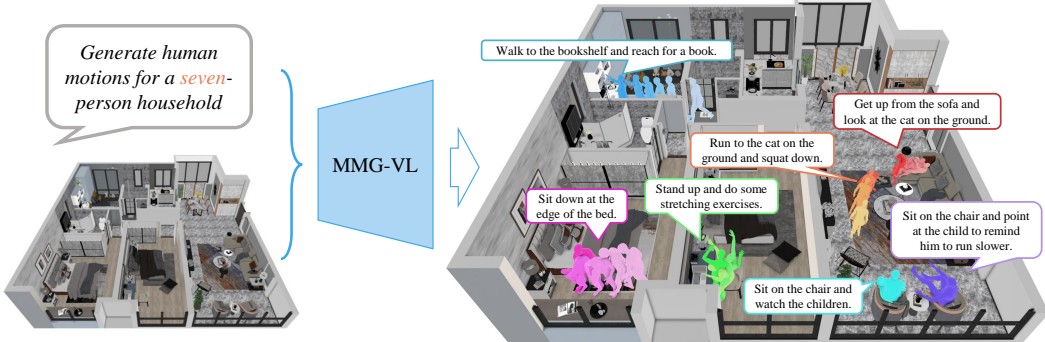

Figure 1: **VL2Motion paradigm:** Given an environmental image and a natural language description, MMG-VL can generate coordinated multi-person motions that interacts naturally with the environment.

## 1 Introduction

At the forefront of Embodied Intelligence research, generating realistic and contextually appropriate 3D human motion is a key technology for achieving natural and immersive experiences, with wide applications in fields such as Human-Computer Interaction (HCI) and Virtual Reality (VR). As the boundaries between virtual environments and the physical world become increasingly blurred, to produce highly realistic motions, systems need to accurately interpret the environment and use this information to generate motions that are physically plausible and contextually appropriate. Visual

perception plays a foundational role in this process, providing the system with key information about the spatial layout, object positions, and dynamic changes in the environment, which directly informs the motion generation process. In multi-person scenarios, the system must also consider the spatial relationships between individuals to ensure that the generated motions are reasonable and coordinated in terms of position and dynamics, ultimately achieving consistency and coherence.

However, most existing human pose generation methods still rely heavily on text or initial pose inputs, primarily encompassing text-to-motion (Ma et al., 2022; Guo et al., 2023; Zhang et al., 2023b; Wang et al., 2022; Athanasiou et al., 2022), action-to-motion (Petrovich et al., 2021; Xu et al., 2023), or a combination of both (Tevet et al., 2023; Jiang et al., 2024; Sun et al., 2024). These methods have significant limitations in dealing with complex environments and integrating multimodal information. Firstly, methods (Wang et al., 2024b; Liang et al., 2024; Chi et al., 2024; Wang et al., 2024a; Mengyi Shan, 2024) that rely on text or initial pose inputs often fail to fully capture the rich semantic information and dynamic changes present in complex real-world environments. Secondly, most existing studies (Tevet et al., 2023; Sun et al., 2024) primarily focus on single-person motion generation, which is insufficient to meet the real-world demands of multi-person scenarios. This limitation is particularly evident in scenarios involving more than two people, undermining the realism and overall performance of motion generation and hindering real-world applications.

To address these challenges, we propose the VL2Motion paradigm for human motion generation, as shown in Figure 1. This paradigm integrates motion descriptions with environmental visual input, leveraging deep multimodal information fusion to generate highly realistic 3D human motion that aligns with real-world semantic logic. By incorporating visual input, VL2Motion enables the system to accurately interpret spatial layouts, environmental details, and the relationships between multiple individuals. Additionally, through the inherent 3D spatial recognition and commonsense constraints within the visual semantics, the generated motions are ensured to be natural and contextually appropriate in complex scenes. This framework utilizes a two-stage pipeline structure, as shown in Figure 2. In the first stage, Vision-Language Auxiliary Instruction (VLAI) module are employed to fuse multimodal input information, transforming open-world natural language instructions into multi-person motion descriptions that adhere to real-world constraints. In the second stage, Scenario-Interaction Diffusion (SID) module is used to further refine and generate multiple human motions. This two-stage design not only enhances the precision and continuity of motion generation but also ensures the coordination and overall consistency of multi-person motion generation, enabling the system to produce realistic and plausible multi-person motions. Additionally, we have constructed and released a complementary dataset HumanVL for VL2Motion. This dataset includes 584 multi-room household images and 35,622 human motion samples. The release of this dataset aims not only to advance research and innovation in the field of Embodied Intelligence but also to lay the groundwork for more complex and diverse application scenarios in the future.

To validate the effectiveness of the MMG-VL approach based on the VL2Motion paradigm, we conducted extensive experiments on the HumanML3D (Guo et al., 2022), InternHuman (Liang et al., 2024), and HumanVL datasets. We performed quantitative assessments using both automated metrics and human evaluation criteria, alongside qualitative evaluations through human judgment. The experimental results demonstrate that, compared to the traditional Text2Motion paradigm, VL2Motion exhibits significant unique advantages in real-world scene perception and interaction. Furthermore, MMG-VL is capable of generating realistic multi-person motions in multi-room home scenarios, with the generated motions significantly outperforming state-of-the-art methods in terms of spatial distribution, environmental interaction, and adherence to common-sense constraints.

Our contributions are summarized as follows: **(1) We propose the VL2Motion paradigm for human motion generation and construct a complementary dataset:** We introduce the VL2Motion paradigm and provide a specially designed dataset HumanVL to promote in-depth research and development in environmental understanding and perception, particularly in generating realistic multi-person motions that align with real-world semantics. **(2) We develop an end-to-end 3D human motion generation model, MMG-VL:** We design and implement an end-to-end 3D human motion generation model, MMG-VL, which can generate multi-person motions in multi-room environments, providing an effective solution for generating realistic multi-person scenarios. **(3) We explore a simple yet effective multi-stage motion generation method:** We propose an innovative multi-stage generation method, first using VLAI to transform open-world natural language instructions into multi-person motion instructions constrained by real-world contexts, followed by the use of SID to generate coordinated multi-person motions based on the diffusion model, thereby significantly enhancing the coherence and naturalness of the generated motions.

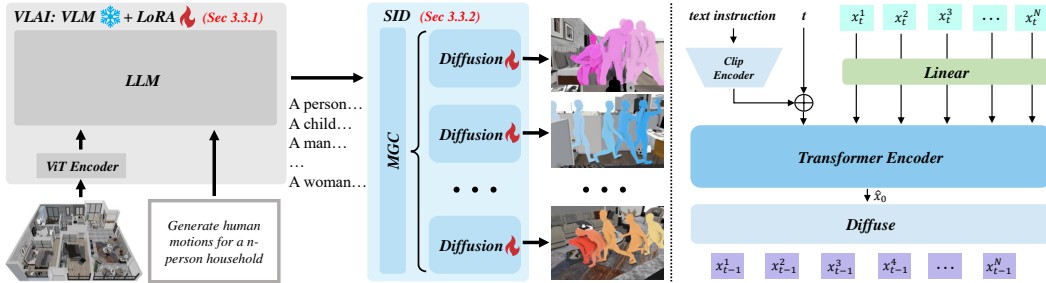

Figure 2: **(Left) Method overview:** We propose the MMG-VL with two key parts: (1) *Vision-Language Auxiliary Instruction (VLAI)*. This part integrates multimodal input information and generates multi-human motion instructions that align with real-world constraints. (2) *Scenario-Interaction Diffusion (SID)*. The SID accurately generates multiple human motions. **(Right) Motion generation based on diffusion models.**

## 2 RELATED WORK

**Human Motion Generation.** In recent years, human motion generation has become a research hotspot due to its broad application prospects in fields such as embodied intelligence, virtual reality, and animation. Numerous studies have focused on generating single-person motion based on various conditional signals, including audio (Ng et al., 2022; 2024), music (Le et al., 2023; Ma et al., 2022; Zhao et al., 2023), action (Petrovich et al., 2021; Tevet et al., 2023; Jiang et al., 2024), and natural language (Ma et al., 2022; Tevet et al., 2023; Guo et al., 2023; Zhang et al., 2023b; Jiang et al., 2024; Sun et al., 2024; Wang et al., 2022; Athanasiou et al., 2022). However, it is regrettable that visual content, a crucial and widely-used information carrier in human life, has not been fully utilized as a conditional input for generating human poses. This omission inevitably leads to a disconnect between the generated motions and real-world environments, significantly limiting their potential in practical applications. Moreover, although some recent studies (Xu et al., 2023; Wang et al., 2024b; Liang et al., 2024; Chi et al., 2024; Wang et al., 2024a; Mengyi Shan, 2024) have begun to explore multi-person human motion generation, most of these efforts remain focused on generating motions for two people, making it difficult to extend to scenarios involving a larger number of individuals. To address these limitations in existing human motion generation methods, we introduce VL2motion, a novel paradigm that extends the Text2Motion framework by incorporating both visual and natural language inputs as conditional signals for generating multi-person human motions.

**Vision Language Models-Guided Diffusion Models.** Vision Language Models (VLM) (Liu et al., 2023b; 2024; 2023a; Zhang et al., 2023c; Dong et al., 2024a;b; Zhang et al., 2024; Chen et al., 2023; 2024b; OpenGVLab, 2024; Bai et al., 2023; OpenAI, 2023b; 2024) have advanced significantly in aligning visual and textual information, driven by breakthroughs in Large Language Models (LLM) (Meta, 2024a;b; Chiang et al., 2023; 01AI, 2024; OpenAI, 2023a). VLMs excel in visual perception and comprehension but still encounter challenges in generative tasks. In parallel, Diffusion Models (Ho et al., 2020; Nichol & Dhariwal, 2021; Rombach et al., 2021) have achieved remarkable success in generation tasks, including human motion synthesis (Zhang et al., 2023b; Tevet et al., 2023; Liang et al., 2024; Chi et al., 2024; Sun et al., 2024), though they struggle with environmental perception and interaction.

Recent work integrates VLMs' perceptual strengths with diffusion models' generative abilities. Mulan (Li et al., 2024) and ConceptLab (Richardson et al., 2024) leverage VLMs to guide diffusion models in text-to-image generation, while DreamArrangement (Chen et al., 2024a) and LVDiffusor (Zeng et al., 2024) apply similar approaches in embodied intelligence tasks. Our research combines these complementary strengths, achieving highly realistic, semantically coherent 3D human motion generation, thus enhancing generative quality and enabling deeper integration of perception and generation.

## 3 METHODOLOGY

Our goal is to generate realistic multi-person human motions based on real-time captured images (which may include multiple rooms) and natural language instructions from the user. The first challenge lies in effectively integrating visual and textual inputs to ensure that the generated human motions adhere to the environmental constraints and are both reasonable and natural. The second challenge is to generate coordinated multi-person motions in one or multiple rooms, ensuring overall consistency and synchronization. To address these challenges, we first introduce the VL2Motion paradigm (see Sec 3.1) and our accompanying dataset, HumanVL (see Sec 3.2). We then present

Table 1: **Dataset comparisons.** We compare our HumanVL dataset with existing human motion datasets. **HSI** refers to Human-Scene Interaction, while **Descriptions** refers to the intermediate low-level motion instructions we preserve in HumanVL.

| Dataset | Natural Language | Image | HSI | Multiple Humans | Multiple Rooms | Descriptions | Motions |
|---|---|---|---|---|---|---|---|
| KIT(Plappert et al., 2016) | ✓ | - | - | - | - | 6278 | 3911 |
| PROX-Q(Hassan et al., 2019) | - | ✓ | ✓ | - | - | - | 60 |
| GTA-IM(Cao et al., 2020) | - | ✓ | ✓ | - | - | - | 119 |
| NTU RGB+D 120(Liu et al., 2020) | - | ✓ | - | - | - | - | 20579 |
| You2Me(Ng et al., 2020) | - | ✓ | - | - | - | - | 42 |
| BABEL(Punnakkal et al., 2021) | ✓ | - | - | - | - | 28055 | 13220 |
| ExPI(Wen et al., 2021) | - | ✓ | - | ✓ | - | - | 115 |
| HUMANISE(Wang et al., 2022) | ✓ | - | ✓ | - | - | 19600 | 19600 |
| HumanML3D(Guo et al., 2022) | ✓ | - | - | - | - | 44970 | 14616 |
| InterHuman(Liang et al., 2024) | ✓ | - | - | ✓ | - | 23337 | 7770 |
| HumanVL(Ours) | ✓ | ✓ | ✓ | ✓ | ✓ | 11874 | 35622 |

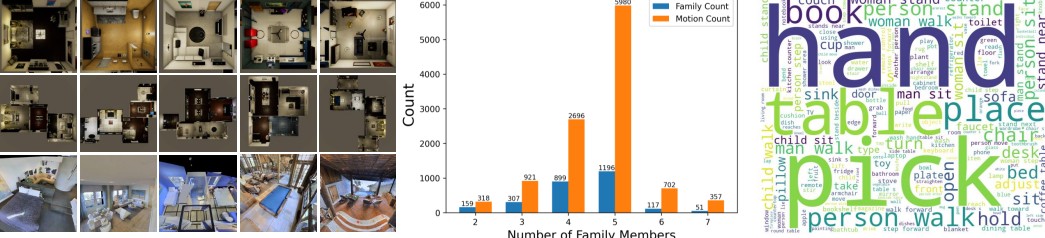

Figure 3: **(Left)** Samples of scenarios in the HumanVL dataset. **(Middle)** Number of households by family size and corresponding motion count in HumanVL. **(Right)** Diverse descriptions in HumanVL.

MMG-VL (see Sec 3.3), an end-to-end framework designed for multi-person, multi-room human motion generation, aimed at producing realistic and well-coordinated human motions.

### 3.1 PRELIMINARY: VL2MOTION

The VL2Motion paradigm aims to generate multi-person motion sequences $x_p^{1:N}$, where $p$ represents the number of individuals, and the motion sequence length is $N$. For each person, the motion at time step $t$, $x_t^i$, is a $J \times D$ dimensional vector, where $J$ is the number of joints, and $D$ is the dimensionality of each joint. The generation of motions is conditioned on multimodal inputs, including natural language descriptions $l$ and visual inputs $v$, which together define the semantics and environmental constraints for the motion generation. The natural language description $l$ provides instructions and objectives for the motion, while the visual input $v$ supplies scene information (such as images of multi-room environments), helping the system understand spatial layouts, object positions, and dynamic constraints within the scene.

Based on these inputs, the system generates motion sequences for $p$ individuals, each sequence containing joint rotation or positional information, ensuring that the motions naturally adapt to the physical constraints of the scene. Under the guidance of both visual and language inputs, the system produces coherent and realistic motions. By deeply integrating natural language $l$ and visual information $v$, the VL2Motion paradigm ensures that the generated multi-person motions not only adhere to the scene requirements but also exhibit high levels of coherence and realism, making them adaptable to complex and dynamic environments.

### 3.2 HUMANVL DATASET.

To advance research in the VL2Motion domain, we present the HumanVL dataset to the academic community, as shown in Figure 3. In contrast to existing datasets, as shown in Table 1, HumanVL is a large-scale 3D multi-person motion dataset based on the VL2Motion paradigm, with a focus on household environments. Each data sample includes both a top-down or bird's-eye view of a household scene, accompanied by text instructions and multi-person motion labels. Additionally, we preserve the intermediate results, linking each individual's motion to the corresponding text instruction, making HumanVL not only valuable for VL2Motion research but also a valuable resource for the Text2Motion community.

To ensure diversity in the dataset, we first collected 10,000 top-down and bird's-eye view images of both single-room and multi-room layouts from four widely used household simulators: iGibson (Li et al., 2021), Virtual-Home (Puig et al., 2018), Matterport3D (Chang et al., 2017), and AI2-THOR (Kolve et al., 2022). From this collection, we meticulously selected 584 high-quality images as the

basis of the dataset. We then designed 2,729 sets of natural language multi-person motion instructions for these images. Notably, in crafting these instructions, we placed a strong emphasis on ensuring the coordination and synchronization of the motions among multiple individuals. This was done to guarantee temporal and spatial coherence in the interactions between people. Furthermore, we carefully considered how the individuals' motions interact with objects and the environment within the scene, ensuring that the instructions respect the physical constraints and logical affordances of the scene. This attention to detail not only enhances the realism of the instructions but also provides robust data for studying collaborative behaviors in complex environments. Each instruction set involves 2 to 7 people, aligning with the typical number of family members in real-world households. Subsequently, we used the MDM (Tevet et al., 2023) to generate 3D human motions corresponding to each set of instructions, ensuring both the reliability and diversity of the motions. The design of the HumanVL dataset not only achieves a high level of complexity and realism but also fills the gap in existing datasets regarding multi-person motion, household scenes, and the generation of 3D motions from natural language descriptions.

### 3.3 MMG-VL: VISION-LANGUAGE DRIVEN DULTI-PERSON MOTION GENERATION

We propose the MMG-VL, an end-to-end framework designed to generate multi-person motion sequences. While we adopt the motion representation format from HumanML3D (Guo et al., 2022), we introduce key extensions to adapt it for the task of motion generation in multi-person scenarios. In MMG-VL, each complete human motion data $M$ consists of $F$ frames and $J = 22$ joints. The motion data format for each individual includes angular velocity and linear velocity of the root joint, local positions, rotation information, joint velocities, and contact signals. Unlike HumanML3D, which only supports single-person motion representation, MMG-VL extends this representation to accommodate multi-person generation. Specifically, at each time step $t$, we generate independent motion sequences $x_t^i$ for each individual $i$. These sequences not only retain the fine-grained motion details from the HumanML3D format, but also ensure that the motions of multiple individuals are generated in a coordinated manner.

The framework is composed of two main components: the first is VLAI, which integrates visual input $v$ and textual input $l$ to generate motion instructions for multiple individuals. The second component is SID, which decomposes the generated instructions into independent motions for each individual. These motions are then generated using a diffusion model to produce the complete motion sequence for each person. This framework ensures that the generated motions are naturally coordinated in complex dynamic scenes, ensuring that each individual's motion adheres to physical constraints while maintaining consistency in multi-person environments.

#### 3.3.1 VLAI: VISION-LANGUAGE AUXILIARY INSTRUCTION

VLAI is a key component of MMG-VL, responsible for integrating visual and linguistic information into low-level textual instructions $c$ to guide subsequent multi-person motion generation. Unlike models that rely solely on textual input, we incorporate visual input $v$ to enhance the system's understanding of the scene, allowing the generated motions to better adapt to physical environmental constraints. The visual input $v$ is processed by a visual encoder to extract critical information such as the spatial layout of the scene and object positions, ensuring that the model fully understands the environment in which the motions will be executed. With the inclusion of visual information, the model can better recognize spatial constraints and dynamic feasibility. For instance, if the scene is identified as a bedroom, the model will automatically avoid generating motions that are incongruous with the environment (e.g., cooking). Simultaneously, the language input $l$ is transformed into high-level semantic representations via a language encoder, capturing the goals and motivations of the motions. The information from these two modalities is fused through a cross-modal attention mechanism, generating a multimodal representation that not only includes the semantic objectives of the motions but also integrates the constraints from the visual scene. This ensures that the generated motions are both contextually appropriate and physically realistic. This fusion process can be formalized as:

$$c = \text{VLAI}(v_{\text{feat}}, l_{\text{feat}})$$

where $v_{\text{feat}}$ and $l_{\text{feat}}$ represent the features extracted by the visual and language encoders, respectively. The final output, $c$, is passed to the subsequent multi-person motion scheduling module, ensuring that the generated motions adhere to environmental constraints while incorporating multimodal information.

### 3.3.2 SID: SCENARIO-INTERACTION DIFFUSION

The main task of the SID is to generate motion sequences for $p$ individuals based on the textual instructions $c$ produced by the VLAI. SID utilizes a diffusion model to generate each individual's motion sequence, ensuring that the generated motions align with the multimodal inputs and that the motions of different individuals are well-coordinated. First, the textual instructions $c$ are decomposed into individual motion guidance signals $c^i$ for each person by the Multi-human Generation Controller (MGC):

$$c^i = f_{\text{MGC}}(c, i)$$

where the function $f_{\text{split}}$ splits the instructions $c$ into independent motion instructions $c^i$ for each individual. The motion generation process for each individual is based on their respective instructions $c^i$, producing the motion sequence $x_t^i$. The diffusion model operates as a Markov noising process. For each individual $i$, the initial motion $x_0^i$ is drawn from a Gaussian distribution:

$$x_0^i \sim \mathcal{N}(0, I)$$

and progressively denoised over time. At each time step $t$, the model generates the motion $x_t^i$ based on the motion from the previous step $x_{t-1}^i$, following the conditional Gaussian distribution:

$$q(x_t^i | x_{t-1}^i) = \mathcal{N}\left(\sqrt{\alpha_t} x_{t-1}^i, (1 - \alpha_t)I\right)$$

where $\alpha_t \in (0, 1)$ are hyperparameters controlling the noise level at each step. The generated motion sequence becomes progressively less noisy as $t$ increases. At each step, the current motion $x_t^i$ is computed using the diffusion model $G$ with guidance from the instructions $c^i$:

$$x_t^i = G(x_{t-1}^i, t, c^i)$$

This iterative process ensures that the generated motion aligns with the individual's guidance while reducing noise over time. Importantly, a noise control mechanism ensures that the generated motions maintain scene consistency and diversity. The final motion sequence is generated by recursively removing noise from the initial random motion. The complete motion sequence $x_t^i$ at each step is a result of the following iterative process:

$$x_t^i = \sqrt{\alpha_t} x_0^i + \sqrt{1 - \alpha_t} \epsilon$$

where $\epsilon \sim \mathcal{N}(0, I)$ represents the Gaussian noise introduced at each step, ensuring the transition from noisy initial motion to the final refined sequence. This continues until the complete motion sequence is generated.

During the generation process, each individual's motion $x_t^i$ is not only guided by their own instructions but is also adjusted to meet the global scene constraints. Ultimately, all individual motions are combined into the final multi-person motion sequence $x_p^{1:N}$, where the motions of each individual adhere to the physical constraints of the scene while remaining coordinated with the motions of others.

## 4 EXPERIMENTS

### 4.1 EXPERIMENTAL SETUP

**Datasets.** The existing human motion datasets lack visual images as inputs and do not include textual task descriptions adapted to daily activities in home environments. Therefore, we contribute a new dataset, HumanVL (see Sec 3.2), which provides a rich set of images depicting home environments and detailed descriptions of everyday tasks in household contexts. It covers daily activities involving multiple individuals and multiple rooms in domestic settings. Additionally, we conduct quantitative comparisons between MMG-VL and existing models on the HumanML3D dataset (Guo et al., 2022) and InterHuman dataset (Liang et al., 2024). HumanML3D is the most widely used text-to-motion dataset, comprising 14,616 single-person motions. InterHuman is the first dataset to feature text annotations for two-person motions. This dataset includes 6,022 motions spanning various categories of human motions and is labeled with 16,756 unique descriptions made up of 5,656 distinct words.

**Evaluation metrics.** We adopt the mainstream quantitative evaluation metrics for human motion generation in the community as Guo et al. (2022), which are as follows: (1) *Frechet Inception Distance* (FID): measures the latent distribution distance between the generated dataset and the real dataset. (2) *R-Precision*: assesses text-motion matching, indicating the probability that the real text appears in the Topk (k=3 in our paper) after sorting. (3) *Diversity*: measures motion diversity in the generated motion dataset. (4) *Multimodality*: gauges diversity within the same text. (5) *Multi-modal distance*: measures the distance between motions and text features.

Table 2: **Quantitative results for single human motion generation on the HumanML3D test set.** All methods use the real motion length from the ground truth. We run all the evaluation 20 times (except MultiModality runs 5 times). **Bold** indicates best result.

| Model | R Precision (Top 3)↑ | FID↓ | Multimodal Dist↓ | Diversity↑ | Multimodality↑ |
|---|---|---|---|---|---|
| Real | 0.797 | 0.002 | 2.974 | 9.503 | - |
| Text2Gesture (Bhattacharya et al., 2021) | 0.345 | 7.664 | 6.030 | 6.409 | - |
| T2M (Guo et al., 2022) | 0.740 | 1.067 | 3.340 | 9.188 | 2.090 |
| MDM (Tevet et al., 2023) | 0.611 | 0.544 | 5.566 | 9.559 | 2.799 |
| MotionDiffuse (Zhang et al., 2022) | 0.782 | 0.630 | 3.113 | 9.410 | 1.553 |
| T2M-GPT (Zhang et al., 2023a) | 0.775 | 0.116 | 3.118 | 9.761 | 1.856 |
| ReMoDiffuse (Zhang et al., 2023b) | 0.795 | 0.103 | 2.974 | 9.018 | 1.795 |
| MotionGPT-13B (Jiang et al., 2024) | - | 0.567 | 3.775 | 9.006 | - |
| MoMask (Guo et al., 2023) | **0.807** | **0.045** | **2.958** | - | 1.241 |
| M2D2M (Chi et al., 2024) | 0.796 | 0.115 | 3.036 | 9.680 | 2.193 |
| MMG-VL (Ours) | 0.653 | 0.521 | 4.988 | **9.790** | **2.967** |

Table 3: **Quantitative results for human-human motion generation on the InterHuman test set.** All methods use the real motion length from the ground truth. We run all the evaluation 20 times (except MultiModality runs 5 times). **Bold** indicates best result.

| Model | R Precision (Top 3)↑ | FID↓ | Multimodal Dist↓ | Diversity↑ | Multimodality↑ |
|---|---|---|---|---|---|
| Real | 0.701 | 0.273 | 3.755 | 7.948 | - |
| TEMOS (Petrovich et al., 2022) | 0.450 | 17.375 | 6.342 | 6.939 | 0.535 |
| T2M (Guo et al., 2022) | 0.464 | 13.769 | 5.731 | 7.046 | 1.387 |
| MDM (Tevet et al., 2023) | 0.339 | 9.167 | 7.125 | 7.602 | 2.355 |
| ComMDM (Shafir et al., 2023) | 0.466 | 7.069 | 6.212 | 7.244 | 1.822 |
| RIG (Tanaka & Fujiwara, 2023) | 0.521 | 6.775 | 5.876 | 7.311 | 2.096 |
| InterGen (Liang et al., 2024) | 0.624 | 5.918 | 5.108 | 7.387 | 2.141 |
| TIM (Wang et al., 2024b) | **0.734** | **4.702** | **3.769** | 7.943 | 1.005 |
| MMG-VL (Ours) | 0.382 | 8.729 | 6.869 | **7.983** | **2.540** |

However, the aforementioned metrics do not fully capture the generative model's ability to perceive and interact with the environment under the VL2Motion paradigm, nor do they assess the coordination and rationality of multi-person motions. To address these gaps, we propose a manual evaluation system that comprehensively measures the rationality, diversity, and real-world applicability of generated motions from a human cognitive perspective. The system includes: (1) *Single-person Quality (SQ)*: evaluates the coherence, naturalness, and physical plausibility of individual motions. (2) *Spatial Distribution (SD)*: assesses the spatial arrangement and movement range of multiple subjects, ensuring reasonable positioning and avoiding overcrowding. (3) *Commonsense Constraints (CC)*: ensures motions align with physical reality and common-sense behavior, such as accounting for object weight. (4) *Environmental Interaction (EI)*: focuses on meaningful interactions with the environment, ensuring motions adapt to specific surroundings. (5) *Multi-person Coordination (MPC)*: measures the synchronization and coordination of motions among multiple subjects, ensuring precise cooperation and avoiding conflicts. (6) *Multi-room Coverage (MRC)*: measures the proportion of rooms engaged by generated motions, indicating effective use of the environment.

**Implementation Details.** In our implementation, MMG-VL consists of two main modules: VLAI and SID, with detailed descriptions provided in Sec 3.3. Specifically, we utilize InternLM-XComposer2.5-7B (Zhang et al., 2024) as the base model for VLAI and MDM (Tevet et al., 2023) as the base model for SID. The training of these two modules is conducted separately. First, we freeze the parameters of the ViT encoder in the VLM and fine-tune the LLM and the projector using the LoRA method (Hu et al., 2022). This stage of training is performed on an Nvidia A100 GPU. Next, we conduct full fine-tuning of the MDM on an Nvidia 2060 Ti GPU using samples from the HumanML3D dataset with lengths exceeding 150 frames, aiming to enhance MDM's ability to generate motion sequences based on long textual descriptions.

## 4.2 QUANTITATIVE RESULTS

**Results on HumanML3D dataset.** In our single human motion generation experiments, we conducted a comprehensive evaluation using the widely recognized HumanML3D dataset. To ensure the fairness and breadth of the evaluation, we systematically compared MMG-VL with 9 state-of-the-art models that have shown strong performance in recent motion generation tasks. The experimental results are detailed in the accompanying Table 2. Although MMG-VL exhibits some performance gaps compared to the current leading model in the key metrics of R Precision (Top 3), FID, and Multimodal Dist, it still demonstrates competitive performance. Notably, MMG-VL slightly out-

Table 4: **Quantitative results for multi-person motion generation on the HumanVL dataset.** We run all the evaluation 20 times. The evaluation was carried out by five PhD candidates, who rated each sample across six dimensions: *Single-person Quality*, *Spatial Distribution*, *Commonsense Constraints*, *Environmental Interaction*, *Multi-person Coordination*, and *Multi-room Coverage*. Each dimension was scored on a scale from 0 to 10, with the final score being the average of all ratings. **Bold** indicates the best result among groups of the same number of people.

| Model | Nums of Human | Single-person Quality↑ | Spatial Distribution↑ | Commonsense Constraints↑ | Environmental Interaction↑ | Multi-person Coordination↑ | Multi-room Coverage↑ |
|---|---|---|---|---|---|---|---|
| MDM | | 4.748 | - | 6.498 | 1.884 | - | - |
| MoMask | 1 | **6.834** | - | 7.576 | 2.746 | - | - |
| MMG-VL (Ours) | | 5.383 | - | **7.625** | **8.202** | - | - |
| InterGen | 2 | **5.820** | 4.865 | 7.660 | 2.253 | **7.847** | 2.410 |
| MMG-VL (Ours) | | 5.429 | **7.462** | **8.873** | **9.220** | 6.452 | **4.197** |
| | 3 | 5.218 | 7.658 | **7.848** | **8.300** | 6.913 | 4.799 |
| | 4 | **5.413** | 8.432 | 7.283 | 8.264 | 6.390 | 4.820 |
| MMG-VL (Ours) | 5 | 5.281 | 8.040 | 6.643 | 7.653 | **7.015** | 5.726 |
| | 6 | 5.108 | 8.219 | 5.390 | 7.209 | 6.583 | 5.819 |
| | 7 | 5.027 | **8.835** | 5.092 | 7.392 | 6.720 | **6.932** |

performs our base model, MDM, across all three metrics, suggesting potential inherent limitations in the MDM architecture. This indicates that future improvements might be achievable by adopting more advanced generative models, potentially narrowing or even surpassing the current performance gap. Moreover, MMG-VL excels in the evaluation of motion diversity and multimodality, achieving the best results to date. This highlights MMG-VL's significant advantages in these crucial dimensions and underscores its considerable potential in enhancing diversity and multimodality in human motion generation.

**Results on InterHuman dataset.** We compared MMG-VL with several state-of-the-art approaches on the InterHuman dataset for human-human motion generation tasks, with the results detailed in the accompanying Table 3. Similar to the findings on the HumanML3D dataset, MMG-VL achieved the best performance in both the Diversity and Multimodality metrics, further validating its significant advantages in generating diversity and multimodal outputs. These results reinforce MMG-VL's leading position in diversity generation and multimodal performance.

**Results on HumanVL dataset.** We conducted an evaluation of multi-person, multi-room human motion generation in domestic scenes using the HumanVL dataset, as shown in Table 4. Due to the unique characteristics of the VL2Motion paradigm, existing human motion generation frameworks do not support visual inputs. Therefore, we compared our approach with models operating under the Text2Motion paradigm. Given that the original textual instructions in HumanVL are abstract directives for generating multi-person motions rather than specific motion descriptions, we employed GPT-4o (OpenAI, 2024) to translate these original instructions into concrete motion descriptions to ensure a fair comparison. These translated descriptions were used as input for the Text2Motion models, while MMG-VL received both the original instructions and corresponding domestic scene images. In the context of single-person motion generation, MMG-VL's output quality was comparable to that of the most advanced models. However, in dual-person motion generation, MMG-VL outperformed the current state-of-the-art model, InterGen, across multiple metrics, including spatial distribution, commonsense constraints, environmental interaction, and multi-room coverage. Notably, in the environmental interaction metric, MMG-VL achieved a score of 9.220, while InterGen scored only 2.253. This stark difference underscores the importance of visual input for environmental awareness and highlights the significant potential of the VL2Motion paradigm in understanding and interacting with realistic environments. Further analysis of MMG-VL's performance in generating motions for three to seven people revealed that as the number of individuals increased, MMG-VL demonstrated increasingly superior performance in spatial distribution and multi-room coverage, while maintaining stable coordination among multiple individuals. This suggests that, thanks to the robust design of the MMG-VL framework, the model can effectively handle the complexity of generating motions for a large number of individuals (more than three) and achieve logical spatial distribution across multiple rooms. However, as the number of individuals increased, MMG-VL's performance in commonsense constraints and environmental interaction showed some decline. We hypothesize that this decline may be due to the increased number of motions generated, which, given the environmental limitations (restricted to the few rooms depicted in the input images), leads to a finite set of interactive objects and feasible motions. Additionally, with a greater number of motions generated, the likelihood of errors increases, which may contribute to the observed decline in commonsense constraint performance.

## 4.3 QUALITATIVE RESULTS

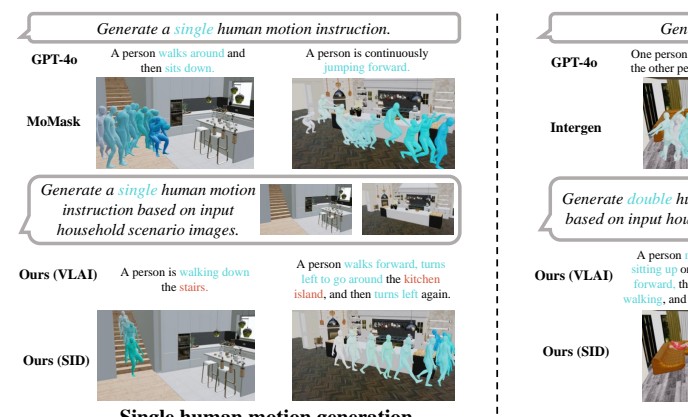

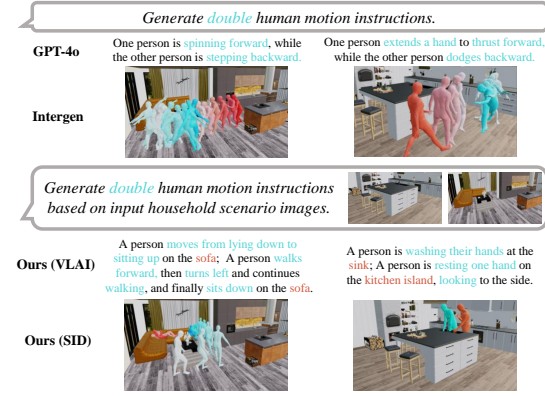

Figure 4: Qualitative comparison with the state-of-the-art single human motion generation method MoMask and the human human motion generation method Intergen.

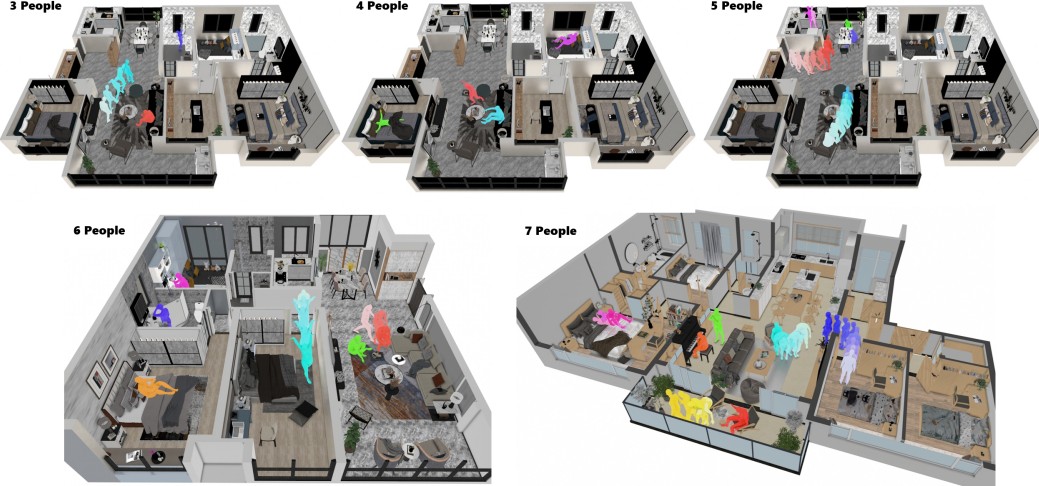

Figure 5: Qualitative results of multi-person motions generated by our MMG-VL in multi-room household scenes.

To validate the effectiveness of MMG-VL, we first conducted a qualitative comparison with the most advanced open-source models in the Text2Motion community: the single-human motion generation model MoMask and the dual-human motion generation model Intergen. Both MoMask and Intergen leverage GPT-4o to generate motion instructions, with the results shown in Figure 4. In the context of single-human motion generation, while MoMask is capable of producing highly realistic and complex movements, it is notably constrained by the limitations of the Text2Motion paradigm, as the LLM-generated motion instructions exhibit significant shortcomings in terms of interaction with the environment. This results in motions that lack authenticity in real-world scenarios. Similarly, in dual-human motion generation, although Intergen is capable of generating motions with strong interactivity between two individuals, the motions tend to be overly generic, making it difficult to demonstrate effective interaction with the surrounding environment. In contrast, MMG-VL excels in both single and dual-human motion generation, demonstrating a high degree of vividness and exhibiting strong environmental interactivity. Furthermore, we present the results of MMG-VL generating multiple human motions within a multi-room environment. As shown in Figure 5, the motions produced by MMG-VL not only display favorable spatial distribution but also closely align with realistic human motions in household scenarios, effectively facilitating interaction with the environment.

## 4.4 ABLATION STUDY

In this section, we investigate the interplay between visual input and natural language input within VL2Motion. As shown in Figure 6, we conducted a qualitative evaluation of MMG-VL using three different input combinations: (A) full text prompts, (B) simple text prompts com-

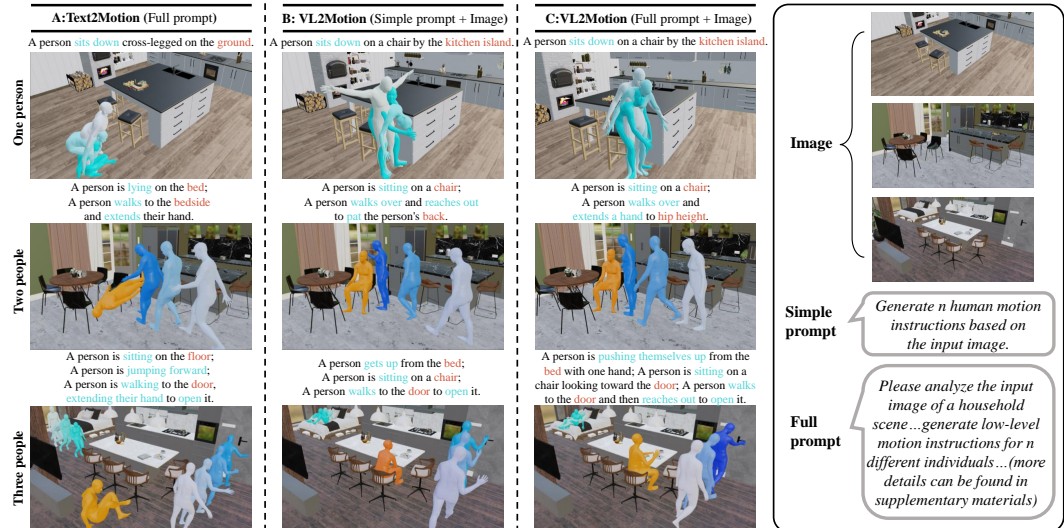

Figure 6: **Ablation study:** we conducted a qualitative evaluation of MMG-VL using three different input combinations: full text prompts, simple text prompts combined with environmental visual input, and full text prompts combined with environmental visual input.

bined with environmental visual input, and (C) full text prompts combined with environmental visual input. When only the text prompt was provided, the human motions generated by MMG-VL failed to effectively interpret environmental information and constraints, resulting in implausible scenarios such as sitting directly on the floor or lying in a room without a bed.

However, with the combination of a simple text prompt and environmental images, the generated human motions demonstrated some degree of interaction with the environment, though they still lacked in detail, such as the naturalness of hand movements. In contrast, when full text prompts were used alongside environmental images, the generated motions were not only realistic and coherent but also adhered to the reasonable constraints of the displayed environment. This highlights the significant advantages of VL2Motion over Text2Motion in terms of understanding and interacting with real-world environments, and underscores that detailed text prompts can substantially enhance the realism of the generated human motions. We also present the quantitative evaluation results of the three combinations in Table 5.

Table 5: For each group size (2 to 7 individuals) in the HumanVL dataset, we selected 3 demos, evaluated using manual metrics, and calculated the average rounded to two decimal places.

|   | SQ | SD | CC | EI | MPC | MRC |
|---|------|------|------|------|------|------|
| A | 5.23 | 8.16 | 4.48 | 3.59 | 6.24 | 4.81 |
| B | 5.18 | 7.89 | 6.14 | 7.47 | 5.22 | 4.39 |
| C | **5.66** | **8.35** | **6.88** | **8.03** | **6.62** | **5.70** |

## 5 CONCLUSION AND LIMITATIONS

**Conclusion.** In this paper, we introduce the VL2Motion paradigm for the first time, aimed at generating realistic 3D human motion that aligns with real-world scenarios by combining environmental visual input and natural language instructions. Additionally, we provide the accompanying 3D human motion dataset, HumanVL. Building on this foundation, we propose MMG-VL, an end-to-end multi-person 3D motion generation method that achieves the generation of multiple human motions interacting naturally with the environment in various rooms of a home setting, while adhering to common-sense principles and maintaining good spatial distribution. We hope our research will offer new insights and inspiration for generating 3D motion in multi-person and complex scenarios.

**Limitations.** Our MMG-VL serves as the first VL2Motion paradigm model in the field, achieving significant advancements in generating human motion for multiple individuals across various rooms, thereby facilitating realistic motion generation and natural interaction with real environments. However, this model still has several limitations. Firstly, despite harnessing the powerful capabilities of VLMs, we have not yet realized scalable multi-human motion generation in the context of generative modeling, which limits the potential for deeper interactions among generated multiple individuals. Secondly, our approach is restricted to generating combinations of two to three human motions, failing to support more complex motion sequences, which affects the model's adaptability in intricate scenarios.

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

# A  APPENDIX

We show our full textual prompt in MMG-VL in Figure 7.

---

**[Full Prompt]**

Please analyze the input image of a household scene, which may be an overhead view of a single room, multiple rooms, or a high-angle shot. Based on the image content, generate low-level motion instructions for 2-7 different individuals in English. Each motion instruction should be a clear sequence of motions without any descriptive statements.

**Requirements:**

Each person should have no more than two motions.

The motion instructions must be brief and concise, specifying body movements, positions, and interactions with objects (e.g., "A man walk forward and use the right hand to pull open the curtain." "A woman sit down and hold the cup with both hands"). Each complete motion sequence should be short and clear.

Ensure that the motions are feasible within the scene and that the individuals' motions do not conflict with each other.

While individuals can perform separate tasks, there should also be some motions that appear interactive (e.g., one person is sitting on a chair, using the right hand to hold chopsticks and eat; another person steps forward to the table and uses the right hand to place the food in his hand onto the table).

The semantic information in the motions must strictly match the image content, with no reference to scenes or objects not present in the image, and must align with common activities in the scene.

Use clear subject identifiers in the motion instructions, such as "a man", "a woman", "a child", "a person" or other specific identities, to clearly indicate each person's motions. Make sure each motion sequence is brief, simple, and feasible for 3D human motion generation.

The output must strictly follow the specified format and include no additional information.

**Output Format Requirements:**

Please output all the motion sequences in English as a single string, with the sequences for different people separated by semicolons.

Within each motion sequence, motions should be separated by commas.

The output must contain only the motion sequences for the exact number of people specified in the task.

Do not include any extra information, labels, or text outside the specified motion sequences.

---

Figure 7: Full textual prompt in MMG-VL.

