# OpenReview forum: "MMG-VL: A Vision-Language Driven Approach for Multi-Person Motion Generation"
_ICLR.cc/2025/Conference — ICLR 2025 Conference Withdrawn Submission_

### Official Review · Reviewer_SRSP · 2024-11-01

**Soundness:** 3
**Presentation:** 3
**Contribution:** 3
**Rating:** 6
**Confidence:** 3

**Summary:**

The paper focus on the multi-person scenarios, combines natural language instruction and environmental visual inputs to generate realistic 3D human motions. The paper propose a generation approach driven by vision and language for generating multi-person human motion in multi-room home scenarios. It firstly use vision-language auxiliary instruction to generate motion instructions align with real-world constraints, then it use scenario interaction diffusion to generate human motions. Moreover, it also provide a dataset, contains multi-room household images.

**Strengths:**

1. The paper explore a challenge but valuable task, generating multi-person interaction motions.
2. The writing is well and contributions is enough, including proposed dataset and network module designs.
3. In the experimental section, the comparison with other methods and the visualization results are sufficient

**Weaknesses:**

1. The analysis and discussion of the dataset in the paper are insufficient. Maybe you can provide more details such as data distribution, annotation process, or specific dataset statistics.

**Questions:**

1. In the dataset, although some physical rules are considered in the instructions, do the interactions between people, people and objects, and people and scenes have physical constraints when generating motions for multiple people?
2. How to ensure that the instructions respect the physical constraints and logical affordances of the scene in dataset, can you show some examples?
3. Does the model incorporate constraints for foot-ground contact or collision avoidance between individuals? If so, how are these implemented within the generation process?
4. One suggestion, a better way to show you generation results with a video.

---

### Official Review · Reviewer_Y1A7 · 2024-11-02

**Soundness:** 2
**Presentation:** 2
**Contribution:** 2
**Rating:** 3
**Confidence:** 4

**Summary:**

This paper introduces a new paradigm for motion generation that integrates natural language instructions with environmental visual inputs to produce 3D human motion. The authors present a two-stage pipeline, consisting of instruction generation followed by motion generation, to achieve this. Additionally, they provide a dataset featuring multi-room, multi-human motion samples.

**Strengths:**

1. The established HumanVL dataset encourages further research on multi-human and scene interaction.

**Weaknesses:**

1. There is a lack of discussion and comparison with prior works on language-guided human motion generation in 3D scenes[1,2]. This paper only incorporates 2D scene information into the generation process, which merely informs the model of where to place the motion. The motivation behind the current setting needs further clarification.

2. The proposed model lacks novelty in design, as it merely combines existing components. It uses a VL module to interpret input descriptions into instructions, which then guide motion generation through an MDM model. However, the model does not incorporate detailed scene information, relying solely on interpreted instructions, which would fail to generate results that align accurately with the scene.

3. The proposed model appears to perform significantly worse than existing methods on R-Precision, FID, and MM-Dist metrics, so it is unclear why the authors claim "it still demonstrates competitive performance."

4. The paper lacks essential details, such as training details and the construction details of the HumanVL dataset. Please refer to the following questions.

[1] Wang et al., HUMANISE: Language-conditioned Human Motion Generation in 3D Scenes. NeurIPS 2022.

[2] Yi et al., Generating Human Interaction Motions in Scenes with Text Control. ECCV 2024.

**Questions:**

- About HumanVL:

1. In the HumanVL dataset, are the language instructions generated manually or produced by an LLM?

2. How is the motion aligned with the 3D scenes?

- About Model:

1. When instructions generated by VLAI relate to different scenarios, the model can generate distinct motions for different individuals. However, how does the model handle this if two instructions refer to the same scenario and overlap spatially?

2. Since the model does not receive 3D scene information as input, how does it achieve the physically plausible results shown in your figures?

- Other comments:

1. The notation $x$ in lines 189 and 191 appears inconsistent in the use of superscripts and subscripts.

---

### Official Review · Reviewer_hpvf · 2024-11-04

**Soundness:** 2
**Presentation:** 2
**Contribution:** 2
**Rating:** 3
**Confidence:** 4

**Summary:**

This paper compensates for the lack of multi-person interaction in previous HSI tasks. It constructs a dataset named HumanVL that contains multi-person actions and is aligned with scenes. On this basis, a model called MMG-VL is designed, providing an effective solution for generating realistic multi-person scenarios.

**Strengths:**

The proposed dataset has relatively rich scenes and includes multi-person actions.

**Weaknesses:**

## W1.Insufficient related work review
1. "FreeMotion: A Unified Framework for Number-free Text-to-Motion Synthesis" and "CrowdMoGen: Zero-Shot Text-Driven Collective Motion Generation" both perform text-to-nPerson motion synthesis and utilize large models for text-level design. These two paper should be cited and the differences between them and this paper should be clarified.

## W2.Dataset construction has some problems.
1. Lack of human-human interaction. Although this paper have placed actions of 2 to 7 people in the room, from the content and materials provided, I don't seem to find any interaction actions between people. If there is no interaction between people, then what is the difference between this and single-person actions?
2. MDM is a model for generating single-person actions and does not consider how to interact with the scene. If human body behaviors are constructed through MDM, how does it interact with the scene?
3. Does the dataset only contain images of the scene? If there is only images and no 3D object representation, how to put motion into the scene?

## W3.The model implementation lacks many details.
1. Why leverage a LoRA to fine-tune a VLM? Since the data scale is limited, will the fine-tuning process cause the VLM to overfit to limited action descriptions, resulting in the loss of generalization ability of the VLM?
2. I am worried that if a full-image are used to extract the viusal feature, how can we ensure that the generated actions can interact with the correct object in the scene?
3. What is the difference between the input representation and the representation provided by HumanML3D?
4. How to ensure that there will be no collisions or interpenetrations between the actions of multiple people generated by only using VLM to extract image features?
5. What is the specific fusion process of $v_{feat}$ and $l_{feat}$ at line 265?
6. Line 279 mentioned that the motion generation process for each individual is based on their respective instructions. Then what is the difference from single-person action generation? Why can this paper achieve the generation of actions of more people?
7. From Line 368, this paper conducts full fine-tuning of the MDM. But how to deal with the difference between the fused feature $c$ in this paper and the text feature in MDM?

## W4.Lack of experimental details.
1. It is best to provide an ablation experiment to verify the role of LoRA, such as removing the fine-tuning process and comparing the results.
2. It is better to describe the details of evaluation metrics (SQ, SD, CC, EI, MPC, MRC).
3. There are few qualitative results, and the generation effect is not persuasive.

**Questions:**

Please refer to the weakness

---

### Official Review · Reviewer_GYtZ · 2024-11-04

**Soundness:** 2
**Presentation:** 2
**Contribution:** 2
**Rating:** 5
**Confidence:** 4

**Summary:**

This paper presents a novel challenge of generating multi-human motions in multi-room home environments. The authors have curated the HumanVL dataset, which contains 584 multi-room household images and 35,622 human motion samples. They present the MMG-VL approach, which uses a visual language model (VLM) conditioned on top-down images of scenes to generate detailed motion descriptions for each individual. Then, leveraging the MDM method, they generate motion sequences for each person individually. The authors compare the performance of MMG-VL with prior methods across three datasets: HumanML3D, InterHuman, and the proposed HumanVL.

**Strengths:**

1. The authors introduce a multi-person, multi-room human motion generation dataset, addressing a notable gap in the field.
2. The qualitative figures are of high quality.

**Weaknesses:**

1. **Layout and Formatting**: The layout is too dense, with minimal spacing before section captions. The authors should consider removing some content or relocating parts to the appendix to restore a more standard layout.
2. **Lack of Motivation for Multi-Person Dataset**: The rationale for introducing a multi-person dataset is unclear. The proposed method and sample data do not explicitly model interactions between individuals, which could be addressed with iterative single-person motion generation. In fact, the HumanVL multi-person data generation proposed here is performed by generating each person individually with MDM, without accounting for other individuals, reducing the significance of a multi-person setting. The current multi-person data appears to be merely a spatial combination of multiple single-person motion sequences. It would be more interesting if the authors consider modeling inter-person interactions.
3. **Unclear SID Module Description**: The SID module is not clearly described. In the formulas in Section 3.3.2, it’s unclear how the SID module interacts with the environment; the single motion sequence seems to rely solely on text instruction $c_i$, which doesn’t make sense in this context. The authors should explain how environmental information is incorporated into the SID module.
4. **Lack of Experimental Metric Descriptions**: There is insufficient explanation of how Diversity, Multimodality, and Multi-modal Distance metrics are calculated.
5. **Typos**:
   - Line 230: "Dulti-person" should be "Multi-person"
   - Line 278: "f_{split}" should be "f_{MGC}"

**Questions:**

The paper needs to reconsider the introduction of multi-person scenarios from both the dataset and method design perspectives. It would benefit from reorganizing the content to clearly articulate the motivation.

---

### Official Review · Reviewer_vMhQ · 2024-11-04

**Soundness:** 2
**Presentation:** 2
**Contribution:** 2
**Rating:** 3
**Confidence:** 5

**Summary:**

This paper presents a method for multi-person motion generation in indoor scenes. The proposed method includes two stages, the vision-language auxiliary instruction (VLAI) module and the scenario-interaction diffusion (SID) module. The VLAI module generates multi-human motion prompts from the visual input and the text. The SID module is a human motion generation module to generate human motion for each individual. And this paper builds a multi-human motion dataset, HumanVL. In the experimental section, the proposed method is evaluated on the single-human motion generation and multi-human motion generation.

**Strengths:**

Here, I highlight the strength of this paper:
1. The problem this paper focuses on, multi-human motion generation in indoor scenes, is worthy of study in the field of human motion generation.
2. This paper provides visualizations of the generation results, which are intuitive.
3. This paper is easy to read.

**Weaknesses:**

Although the problem this paper focuses on is valuable, this paper only proposes a simple pipeline based on existing methods, the novelty of the proposed method is limited. Here I highlight my major concerns.

1. The novelty of the proposed method is limited. For the two key modules in the proposed method, the VLAI uses existing ViT encoder and LLM, and the SID uses existing human-motion methods. Thus, the proposed method appears to be a combination of existing methods, and lacks substantial innovation.
2. About the modeling of human-scene interaction and human-human interaction. This paper relies only on LLM for scene and text understanding, and does not explicitly model these interactions. Then the upper limit of the method depends on LLM, and the error output of LLM will also affect the performance of the method.
3. The user study is unscientific. In the user study in this paper, only five PhD candidates took part in the experiment. The small number of people in the user study, the lack of details of the user study, etc., will make people question the user study.
4. The performance on the main metrics for human motion generation is bad, like FID, R Precision. These metrics reflect the quality of the generation.

**Questions:**

1. What is fsplit in L278?

2. From L354 to L362, there is only the definition of the metric, not how the metric is calculated.

3. The input of ViT encoder is 2D image of the scene, how to understand the 3D interaction between humans and scene?

4. In the limitation, the paper claims that the proposed method only can generate two to three human motions, why does the first image use 7 people as an example?

---

### Note · Authors · 2024-11-12

I have read and agree with the venue's withdrawal policy on behalf of myself and my co-authors.